# Maternal Microbiota Modulate a Fragile X-like Syndrome in Offspring Mice

**DOI:** 10.3390/genes13081409

**Published:** 2022-08-08

**Authors:** Bernard J. Varian, Katherine T. Weber, Lily J. Kim, Tony E. Chavarria, Sebastian E. Carrasco, Sureshkumar Muthupalani, Theofilos Poutahidis, Marwa Zafarullah, Reem R. Al Olaby, Mariana Barboza, Kemal Solakyildirim, Carlito Lebrilla, Flora Tassone, Fuqing Wu, Eric J. Alm, Susan E. Erdman

**Affiliations:** 1Division of Comparative Medicine, Massachusetts Institute of Technology, Cambridge, MA 02139, USA; 2Laboratory of Pathology, Faculty of Veterinary Medicine, Aristotle University of Thessaloniki, 54124 Thessaloniki, Greece; 3Department of Biochemistry and Molecular Medicine, School of Medicine, University of California Davis, Sacramento, CA 95817, USA; 4College of Health Sciences, California Northstate University, Rancho Cordova, CA 95670, USA; 5MIND Institute, University of California Davis, Sacramento, CA 95817, USA; 6Department of Biological Engineering, Massachusetts Institute of Technology, Cambridge, MA 02139, USA; 7Center for Infectious Diseases, School of Public Health, The University of Texas Health Science Center, Houston, TX 77030, USA; 8Broad Institute of MIT and Harvard, Cambridge, MA 02142, USA

**Keywords:** microbiome, FXS, FMRP, probiotic, *Lactobacillus reuteri*

## Abstract

Maternal microbial dysbiosis has been implicated in adverse postnatal health conditions in offspring, such as obesity, cancer, and neurological disorders. We observed that the progeny of mice fed a Westernized diet (WD) with low fiber and extra fat exhibited higher frequencies of stereotypy, hyperactivity, cranial features and lower FMRP protein expression, similar to what is typically observed in Fragile X Syndrome (FXS) in humans. We hypothesized that gut dysbiosis and inflammation during pregnancy influenced the prenatal uterine environment, leading to abnormal phenotypes in offspring. We found that oral in utero supplementation with a beneficial anti-inflammatory probiotic microbe, *Lactobacillus reuteri*, was sufficient to inhibit FXS-like phenotypes in offspring mice. Cytokine profiles in the pregnant WD females showed that their circulating levels of pro-inflammatory cytokine interleukin (Il)-17 were increased relative to matched gravid mice and to those given supplementary *L. reuteri* probiotic. To test our hypothesis of prenatal contributions to this neurodevelopmental phenotype, we performed Caesarian (C-section) births using dissimilar foster mothers to eliminate effects of maternal microbiota transferred during vaginal delivery or nursing after birth. We found that foster-reared offspring still displayed a high frequency of these FXS-like features, indicating significant in utero contributions. In contrast, matched foster-reared progeny of *L. reuteri*-treated mothers did not exhibit the FXS-like typical features, supporting a key role for microbiota during pregnancy. Our findings suggest that diet-induced dysbiosis in the prenatal uterine environment is strongly associated with the incidence of this neurological phenotype in progeny but can be alleviated by addressing gut dysbiosis through probiotic supplementation.

## 1. Introduction

The composition of an individual’s gut microbiome has influence over their overall health via the hypothalamic–pituitary–adrenal axis, connecting the brain and the gut [1]. A person’s gut microbiome is first established in utero and is influenced by their mother’s own gut microbiome [2,3,4]. A mother’s microbiome can enter a state of dysbiosis, defined as an “imbalance” in the gut microbial community, when consuming a high-fat diet, such as a Westernized diet (WD) [5,6]. Consequently, maternal microbial dysbiosis causes an inflammatory response in the mother that triggers phenotypic outcomes including obesity, cancer, and neurological disorders in their offspring [7,8]. Supplementation with probiotics that modulate host inflammation and hormones [9,10] has previously been shown to reverse the adverse effects of maternal gut dysbiosis, and in some cases erase the aberrant phenotype altogether. Dysbiosis during gestation has been linked to increased incidences of multigenerational obesity and cancer, through a T helper 17 cell (Th17)-biased immune response that was reversible with in utero supplementation with *Lactobacillus reuteri*.

A cohort of mice born from descendants of mother mice fed a WD, described by Poutahidis et al. (2015), exhibited craniofacial features, hyperactivity, and stereotypy resembling Fragile X Syndrome (FXS) in humans. FXS is a genetic neurodevelopmental disorder (NDD) caused by a trinucleotide expansion (>200 CGG repeats), the hypermethylation of the 5′UTR end of the fragile X messenger ribonucleoprotein 1 (*Fmr1*) gene, and consequent absence of the encoded protein, FMRP [11,12,13,14,15]. FXS is the most common inherited form of intellectual disabilities in humans, affecting about 1 out of every 4000–5000 males and about 1 out of every 6000–8000 females [16,17]. It is also the most common monogenic cause of autism, accounting for approximately 1–6% autistic spectrum disorder (ASD) diagnoses [16,18,19]. Individuals diagnosed with FXS exhibit a wide spectrum of symptoms including epilepsy and seizures [17,20], behavioral abnormalities [17,21], hypersensitivity to sensory stimuli [22], and autistic-like behaviors [19,23,24,25,26]. Many of the psychiatric and neurological symptoms of FXS result from the unchecked activation of a metabotropic glutamate receptor (mGluR5), an excitatory neurotransmitter [27]. Additionally, a decrease in *Fmr1* protein synthesis induces the downregulation of GABA receptor synthesis [8,28], and results in an imbalance of excitatory and inhibitory neurotransmitters in an individual’s brain [29].

In our research mouse colony, the only common factor in the animals exhibiting the Fragile X syndrome-like phenotype (FXSL) was the microbiome of ancestors that occurred after consuming a WD. The relationship between FXS and the gut microbiome is largely unexplored, suggesting that the discovery of a link between maternal gut dysbiosis and FXS development would be novel. To determine if these mice indeed expressed a Fragile X-like phenotype, we measured elements of their physical and behavioral phenotype, as well as brain FMRP expression levels. We found that supplementation of probiotic *L. reuteri* was sufficient to counteract the neurodevelopmental phenotype and FMRP expression in the offspring mice. We further observed that these phenotypes were retained after C-section rederivation with foster moms hosting different microbiota, suggesting a connection between in utero effects of the maternal gut dysbiosis and this interesting neurodevelopmental phenotype in the offspring animals.

## 2. Materials and Methods

### 2.1. Animals

Outbred conventional CD-1 Swiss stock mice (Charles River; Wilmington, MA, USA) were housed and handled in the Association for Assessment and Accreditation of Laboratory Animal Care-accredited facilities with diets, experimental methods, and housing as approved by the Institutional Animal Care and Use Committee. These studies utilized outbred stock mice with robust breeding capacity and absent any transgenic manipulations predisposing to pathology. To test the impact of dietary microbes on progeny, the experimental design was to expose F0 (grandmother) mice to special diets, starting with mating at the age of 8 weeks, as previously described in detail [7]. Special dietary treatment continued until the birth of their pups. Progeny of subsequent generations were later examined to determine health risks. Because of early life morbidities, in some cases, F2 and F3 progeny (grandchildren) were euthanized at 6 months of age or younger according to institutional humane criteria and clinical disease. Euthanasia was performed using carbon dioxide overdose. Animals underwent videotape analyses of behaviors. Tissues were collected upon necropsy and then examined histologically. Each experiment included 5 to 10 animals per sex per treatment group, performed in duplicate, as described in detail below.

### 2.2. Experimental Design

An experimental overview is provided in Figure 1. Mice in the present experiments received ad libitum water and control diet Prolab IsoPro RMH 3000 [PMI Nutrition International, LLC, Arden Hills, MN, USA]. Grandparent outbred CD-1 stock mice were placed on experimental diets during pregnancy: control AIN-76A (Harlan-Teklad; Madison, WI, USA) and a Westernized diet (WD) with a high fat content and low fiber, with substandard levels of vitamins B and D (TD.96096; Harlan-Teklad; Madison, WI, USA), as previously described [7]. Progeny mice in those earlier studies exhibited increased frequency of cancer, obesity, infertility, premature aging, and an uncharacterized neurological syndrome [7]. The present study focused on offspring with a syndrome of behavioral atypia and features resembling FXS in humans. Those progeny mice were characterized using videotape analyses of behaviors and post-mortem evaluation of tissues.

Subsequently, to test our microbe-centric inflammation-driven hypothesis, 12 pregnant female mice descending from WD-fed CD-1 mother mice in earlier experiments were randomly subdivided into a group of 6 pregnant female mice with half receiving *L. reuteri* ATCC-PTA-6475 in their drinking water throughout pregnancy, as described elsewhere [7]. The other 6 pregnant females received regular drinking water and served as controls. Blood was collected at day 14 of pregnancy to assess the inflammatory cytokines profile.

Finally, we used Caesarian (C-section) rederivation to test our hypothesis that in utero microbial events were driving aberrant behavioral phenotypes in offspring. We used pregnant female mice that descended from WD-fed CD-1 lineage from our earlier experiments and received either supplemental *L. reuteri* (*n* = 6) or control water (*n* = 6). Mice at e20 underwent C-section rederivation using CD-1 foster mothers with dissimilar microbiota to minimize vaginal or post-partum transfer of microbiota to progeny.

### 2.3. Microbial Treatments

Subsets of CD-1 mice who had descended from mice treated with a WD in utero subsequently received an anti-inflammatory strain of *L. reuteri* ATCC-PTA-6475 in their drinking water, cultivated as described elsewhere [7]. Live organisms were supplied at a starting dosage of 3.5 × 10^5^ organisms/mouse/day in drinking water. Control mice received regular drinking water. Fresh drinking water for both groups of animals was replaced twice weekly throughout the experiments.

### 2.4. Phenotyping Mice

Dysmorphia of head and ears are characteristics of FXS in humans. To examine these characteristic features in our mice, we tested ear pinnae morphology by measuring the height and width of each ear and estimating the effective diameter (square root of height * width), as previously described [30,31]. Cranio-metric analyses used Radiographic landmarks as in: X-ray Annotation Mouse Atlas, IMPC, 2021 and Cranial bone morphometric study among mouse strains (nih.gov), using 5–10 mice per group. To examine other classical features of FXS in humans including hyperactivity and head-bobbing stereotypy, we examined videotapes of animals with or without in utero supplementation of *L. reuteri*. Six animals per group were measured in 30 s intervals in home cages under standardized video conditions. Behavioral scores were compiled as the number of times the cage midline was crossed for hyperactivity, and the number of times the rostrum crossed a horizontal plane for head bobbing stereotypy. Finally, after euthanasia and necropsy, formalin-fixed paraffin-embedded head and brain tissues were examined by two board-certified veterinary pathologists (SEC and SM) for possible explanations of FXS-like phenotypes. Tissues were processed for histopathology, as described below.

### 2.5. Caesarian Rederivation of Offspring

Pregnant female CD-1 mice (*n* = 12) who experienced the various treatments explained above underwent terminal sterile C-section rederivation procedures of e20 offspring to test post-partum effects of microbiota. All timed pregnant foster moms had microbiota that were different from donor mice but similar to one another to reduce experimental variables.

### 2.6. Histopathology

For histologic evaluation, formalin-fixed brains were embedded in paraffin, cut at 5 μm, and stained with hematoxylin and eosin (H&E). The mouse heads were fixed in 10% neutral buffered formalin for 48 h, followed by immersion in a decalcification solution for 72 h (Cal-RiteTM; Kalamazoo, MI, USA), and then sections of ears were processed for routine H&E staining. Additionally, tissues of representative areas of brain were compared side-by-side between mice exhibiting Fragile-X-Syndrome-like phenotype (FXSL), clinically normal FXSL-prone mice that had been treated in utero with *L. reuteri*, and clinically normal untreated sham control CD-1 animals. Samples were analyzed by two board-certified veterinary pathologists (SEC and SM) blinded to sample identity.

### 2.7. FMRP Expression

Snap-frozen brain tissues derived from treatment groups were homogenized with a tip sonicator (Q-Sonica; Newton, CT, USA) in a homogenization buffer containing 20 mM HEPES buffer (Thermo Fisher Scientific; Waltham, MA, USA), 0.25 M sucrose (MiliporeSigma; Burlington, MA, USA), and 1X protease inhibitor cocktail (MiliporeSigma; Burlington, MA, USA). Homogenates were centrifuged at 200,000 rpm for 45 min at 4 °C and cytoplasmic fractions (supernatant) were collected and quantified using Bradford assay (BioRad Laboratories, Inc.; Hercules, CA, USA). Overall, 10 µg of protein lysate from each sample was loaded on a 4–12% Bis-Tris gels (BioRad Laboratories, Inc.; Hercules, CA, USA) and run at 100 V for 60 min and 130 V for 60 min. Resolved proteins were then transferred onto nitrocellulose membranes using the Trans-Blot Turbo transfer system (BioRad Laboratories, Inc.; Hercules, CA, USA) at 25 V, 1.0 A for 30 min. Membranes were stained with Ponceau to test for transfer efficiency, and blocked with 3% milk for 1 h at room temperature followed by overnight incubation at 4 °C, with 1:2000 diluted FMRP primary antibodies (ab 17722, abcam; Cambridge, MA, USA). Membranes were then washed in 1X-TBS and incubated with HRP-linked secondary antibody diluted 1:2000 (Catalog#7074, cell signaling technologies; Danvers, MA, USA) for 1 h at room temperature. Bands were then visualized using Chemiluminescent substrate, Super Signal West Dura (Thermo Fisher Scientific; Waltham, MA, USA). Densitometry analysis of bands for relative protein quantification was performed using the Alpha Innotech Gel Imaging System (Cambridge Scientific; Watertown, MA, USA). Experiments were carried out in triplicate.

### 2.8. Inflammatory Cytokines

Serum cytokine protein levels of 8 animals per group were analyzed using the mouse cytokine discovery assay system (Eve Technologies; Calgary, AB, Canada), diluted 1:1 and according to the vendor’s protocol. To test systemic levels of Il-17A in pregnant mice, we used whole blood collected by terminal cardiac puncture at e14 and then diluted 1:1, to assess circulating Il-17A levels using ELISA at Eve Technologies (Calgary, AB, Canada). Samples were analyzed in duplicate.

### 2.9. Statistics

Mann–Whitney U test was used for most analyses unless otherwise stated. Kruskal–Wallis analysis of variance by ranks followed by the post hoc analysis using Dunn’s multiple comparison test was used for head dysmorphia comparisons. Samples for FMRP were run in three independent experiments, and then analyzed using two-tailed Student’s *t*-tests, with a *p*-value < 0.05 considered statistically significant. Statistical analysis of inflammatory cytokines was performed using two-tailed Student’s *t*-test; error bars demonstrate the standard error (SE) of the mean; a *p*-value < 0.05 was considered statistically significant.

## 3. Results

### 3.1. Offspring Mice Exhibit a Fragile X Syndrome (FXS)-like Phenotype including Elongated Head, Enlarged Ears, Head Bobbing, and Hyperactivity

We observed that descendants of mice fed a Westernized diet (WD), as described by Poutahidis et al. (2015) [7], exhibited higher frequencies of stereotypy, hyperactivity, and cranial features, similar to the phenotypic expression of Fragile X syndrome (FXS) in humans. For this study, Figure 1 shows an overview of the experimental design. Recognizing from earlier work that the phenotype in mice fed a WD was transmissible to offspring using fecal microbiome transplant alone into germ-free pregnant mice fed a standard diet [7], we surmised that the effects of deleterious microbiota contributed to the observed pathology in offspring animals.

To further characterize the atypical features, we measured the anatomy and behaviors in the progeny mice descended from the cohort of mice fed the WD. Thus, we compared features potentially linked with embryonal migration of neural crest cells, as postulated in human subjects with FXS, between control CD-1 mice and those descended from WD-fed mothers. We found trends between the Fragile X syndrome-like phenotype (FXSL) mice and sham control mice in ear pinnae morphology using the methods described in Anbuhl et al. (2017) by measuring the height and the width of each ear in different treatment groups [30]. Likewise, FXSL skulls and control skulls displayed some differences in anterior–posterior (AP) skull lengths (Figure 2) [32,33]. Upon videotape analysis, we discovered significant differences between behavioral presentations (Figure 3). In sum, these phenotypic differences were significant when examined in the progeny of different treatment groups (Figure 4).

To further probe phenotypes and potential biomarkers within mouse brains, formalin-fixed paraffin-embedded brain tissues were examined by two board-certified veterinary pathologists who found no evidence of brain, structural, or inflammatory abnormalities in the different treatment groups. There was no evidence of necrotizing polyarteritis of blood vessels, no evidence of intranuclear inclusions in neurons and astrocytes, and Purkinje cells were found to be within normal limits in all animals. In summation, the histopathology does not explain the clinical presentation seen in the FXSL mice.

### 3.2. In Utero L. reuteri Supplementation in Drinking Water Counteracts FXS-like Phenotype in Offspring Mice

Results from Poutahidis et al. (2015) showed that, in germ-free mothers that received fecal microbiome transplantations, diet-induced microbiota alone were sufficient to re-create the multigenerational disease phenotypes in progeny mice [7]. To test this hypothesis that in utero exposures to microbes alone are sufficient to influence the neurodevelopmental phenotypes of progeny animals, we used supplementation with beneficial microbiota, the probiotic *Lactobacillus reuteri*, during pregnancy. We examined whether *L. reuteri* could counteract the FXSL phenotype (Figure 1) and found that supplementation with this probiotic ameliorated the effects of the WD and counteracted the FXSL phenotype in progeny (Figure 2, Figure 3 and Figure 4).

### 3.3. Pregnant Mice Supplemented with L. reuteri Exhibit Lower Circulating Levels of Pro-Inflammatory Cytokine Interleukin -17A

It was previously shown that supplementation of an anti-inflammatory strain of *L. reuteri* leads to lower levels of Il-17A in mouse models [34]. We hypothesized that a dysbiosis-triggered Il-17A-mediated immune response during pregnancy could also be linked to increased instances of FXSL. We found that the mothers descended from a WD lineage had higher levels of pro-inflammatory cytokine Il-17A (Figure 5), suggesting that dysbiotic maternal microbiota induced an inflamed in utero environment predisposing offspring to atypical neurodevelopmental phenotypes. To further test the possibility that in utero effects of microbiota may be important in the FXSL phenotype, we interrupted the transmission of microbes between generations using C-section rederivation and dissimilar foster mothers.

### 3.4. Fragile X-like Phenotype in Mice Is Retained after Caesarian-Section Rederivation

To determine if the FXSL phenotype was attributable to *in utero* effects of microbiota, a C-section rederivation was performed on the mice to reduce post-partum effects of pathogenic microbiota. We found the neurodevelopmental disorder was maintained despite the C-section and subsequent cross-fostering to different mothers (Figure 1). In contrast, matched foster-reared progeny of *L. reuteri*-treated mothers did not exhibit the typical features seen in humans with FXS, supporting a key role for microbiota during pregnancy (Figure 4). Together, these findings further support our hypothesis that the cause of the FXSL phenotype was in utero effects of gut microbiome dysbiosis.

### 3.5. In Utero Microbial Rescue with L. reuteri Blunts Methylation of FMRP

A clear marker of FXS in humans and animal models is the decrease in *Fmr1* protein (FMRP) production following a trinucleotide CGG expansion and hypermethylation of the *Fmr1* gene [12]. Therefore, we quantified FMRP levels in the brains of FXSL mice to determine if there was a difference in FMRP expression, and found significantly lower FMRP levels in the brains of the FXSL mice compared with their clinically normal *L. reuteri*-treated counterparts (Figure 6).

## 4. Discussion

The research described by Poutahidis et al. (2015) revealed multigenerational effects of mothers being fed a Westernized diet (WD), which seemingly impacted the gut microbiota and systemic immunity [7]. In those studies, the progeny of CD-1 stock mice fed a WD were predisposed to premature aging and cancer, despite being fed a normal diet themselves. In addition, the offspring developed a neurologic syndrome that could be transmitted to naïve mice via a gut microbiota-transplant and could be mitigated by a dietary probiotic. In this present study, we worked to further characterize the neurologic syndrome, and found elongated heads, large ears, hyperactivity, and repeated head-bobbing behaviors, largely in male progeny that had not undergone in utero probiotic treatment. These atypical behaviors and features remarkably resembled those observed in human patients with Fragile X Syndrome (FXS) and in *Fmr1*-knockout (KO) mice, which lack normal *Fmr1* protein (FMRP) [14]. Interestingly, an association between the reduction in FMRP expression and the degree of the severity of neurologic phenotypes, including autism spectrum disorder, schizophrenia, bipolar disorder, and major depressive disorder, has been reported in several studies [35,36,37]. This motivated us to examine FMRP levels, which were found to be significantly higher in progeny mice following in utero probiotic treatment. Upon further testing, we discovered that this neurologic syndrome in outbred Swiss mouse offspring was retained following C-section rederivation with rearing by unrelated foster moms. While the *Fmr1*-KO mouse model presents with an altered gut microbiome [38,39], the finding of a microbially driven FXS-like phenotype in animals not otherwise genetically prone would be a novel discovery.

In recent years, there has been considerable interest in the role of in utero infection in the health outcomes of progeny [9,40,41]. Fast-food-style diets in mothers leads to gut dysbiosis that cultivates a uterine environment contributing to increased risk of children developing autism [22]. Indeed, a dysregulated microbiome after a WD has been shown in animal models to lead to neurodevelopmental conditions [42,43,44] that also respond favorably to probiotics [40]. Our own future studies will probe possible interactions between probiotics and candidate microbial pathogens during dysbiosis conditions in utero. There is growing evidence that such shifts in maternal gut microbiota can contribute to autism spectrum disorder [9] aligned with maternal Th-17 inflammatory and stress hormone responses [45]. In the present study, we found that pathology in C-section-derived progeny was inhibited by in utero treatment with probiotic *L. reuteri* coincident with lowered levels of inflammatory cytokine Il-17A. Based on these data, the transfer of microbes at birth was unnecessary, implicating pathogenic microbiota and Il-17A in etiopathogenesis [46]. Similar beneficial outcomes after probiotic treatment were previously inversely linked with stress hormone corticosterone dependent upon oxytocin levels in adult animals [45].

The present original and novel research shows lower FMRP expression linked with in utero exposure to microbiota, perhaps of epigenetic origins (see our companion paper in this same Special Issue [47]), in a neurodevelopmental phenotype arising after in utero microbial perturbations [7]. The reduction in FMRP likely contributes to the symptoms seen in FXS patients, specifically because FMRP is an RNA-binding protein that acts as a translational regulator of neuronal mRNAs of many messages that affect synaptic plasticity, connectivity, and memory in the central nervous system [48]. The absence of FMRP leads to increased glutamate signaling and downregulation of GABA_A_ pathways, causing an imbalance of excitatory and inhibitory neurotransmitters in the brain [11] that likely contributes to the hyperactivity typically seen in FXS. It would be interesting in future studies to correlate FMRP expression levels with individual mouse behaviors, particularly considering the variability in FMRP expression seen among mice in this study. In humans, FMRP expression is reported to vary among clinically normal and FXS subjects [49,50,51,52]. Investigating other genetic markers of FXS in these FXS-like mice would also help to further characterize this model.

In this model, C-section rederivation was performed to temporally isolate effects and test our hypothesis that pivotal microbial events are happening in utero—rather than post-partum—therein modulating the neurological phenotypes. We hypothesize that microbial stressors during pregnancy lead to elevations in inflammatory cytokines that directly or indirectly contribute to stress-induced epigenetic changes in developing embryos [10]. The present study measured inflammatory cytokines in this mouse model at e14, whereas suspected in utero events contributing to FXS in humans happen during the first trimester nearer to 11 weeks of human pregnancy [14]. Specific gut microbiota that triggered pathogenic and potential epigenetic changes are being investigated separately (see our companion paper in this Special Issue [47]).

We found a novel in utero association with maternal exposures to microbiota abolishing FXS-like features in offspring mice. Several other recent studies have shed light on the role of intestinal microorganisms in epigenetic alterations [46,53,54,55]. In those studies, an imbalanced gut microbiome was frequently observed in children with neurodevelopmental disorders [53,54], although our present findings indicated specifically maternal microbiota and in utero events led to the atypical phenotypes in subsequent generations. In either event, alterations in epigenetic processes arising early in life can lead to several neuropsychiatric conditions, including FXS and ASD [56,57,58]. Separately we are examining any potential interaction between the *L. reuteri* intake and changes in the methylomic profiles correlated with the observed phenotypes.

In summary, we present evidence that in utero effects of a ‘stress microbiome’ leads to a neurological phenotype in offspring animals, similar to those observed in FXS, an outcome that is neutralized by feeding of probiotic *Lactobacillus reuteri* during pregnancy. This would also be of interest for comparison with an analogous *Fmr1*-KO mouse model. Targeted infection studies are underway to test candidate pathogens to recreate FXS-like syndromes in mice; nonetheless, targeted microbe strategy using probiotic *L. reuteri* counteracted the development of FXS-like symptoms when introduced during pregnancy, a protective effect that was sustained after C-section rederivation. We conclude that administration of *L. reuteri* during pregnancy with a WD effectively neutralized the symptoms of the neurological syndrome after the birth of infants, raising the possibility of similar therapeutic strategies in humans.

## Figures and Tables

**Figure 1 genes-13-01409-f001:**
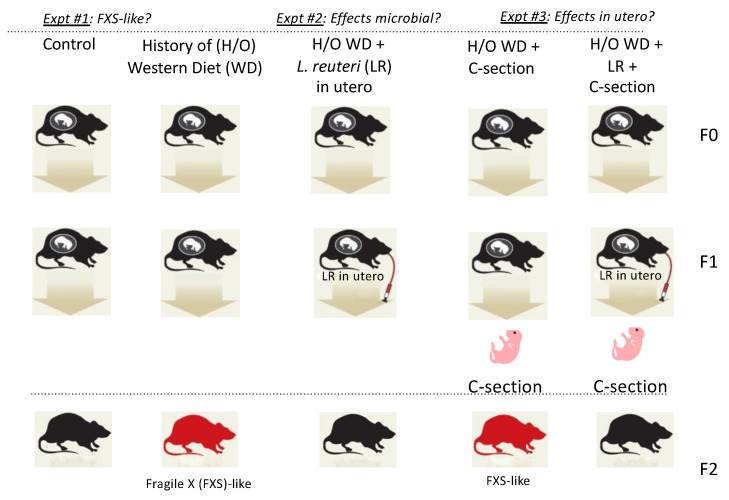
Experimental overview. The present study focused upon offspring from an earlier multigenerational study that spontaneously exhibited a syndrome of behavioral atypia and features resembling phenotypic features of FXS in humans. These mice were subsequently characterized using morphometrics, videotape analyses of behaviors, and post-mortem evaluation of tissues, displaying a neurodevelopmental phenotype with Fragile X-like features. To test a microbe-driven hypothesis, pregnant mice with a history of (h/o) WD (*n* = 12) were then randomly subdivided with half receiving a probiotic *L. reuteri* (LR) ATCC-PTA-6475 in their drinking water. These animals underwent testing of inflammatory cytokines, and their progeny was examined for FXS-like phenotypes and FMRP expression levels. Finally, pregnant mice underwent Caesarian (C-section) rederivation to test our hypothesis that in utero microbial events rather than post-partum microbes were leading aberrant behavioral phenotypes in offspring.

**Figure 2 genes-13-01409-f002:**
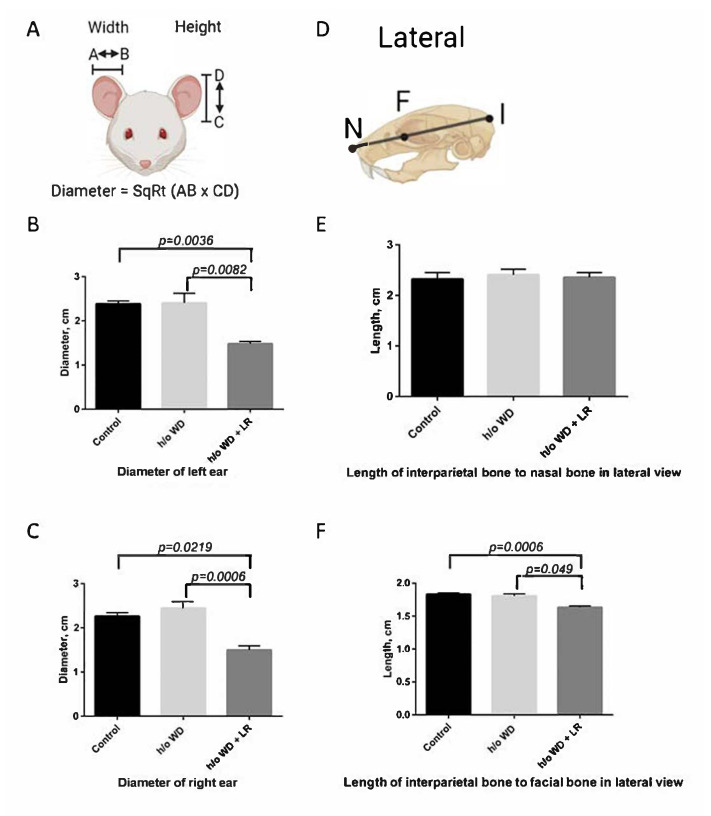
Measuring dysmorphia of head and ears. Misshaped head and ears are a characteristic phenotype of FXS in humans. We tested ear pinnae morphology by measuring the height and the width of each ear and estimating the effective diameter to examine the presence of this phenotype in our mice. Analysis of the control group (*n* = 7), the history of (h/o) WD group (*n* = 5), and the h/o WD + LR group (*n* = 8) showed significant differences in the ear size and the skull width (*p* < 0.05).

**Figure 3 genes-13-01409-f003:**
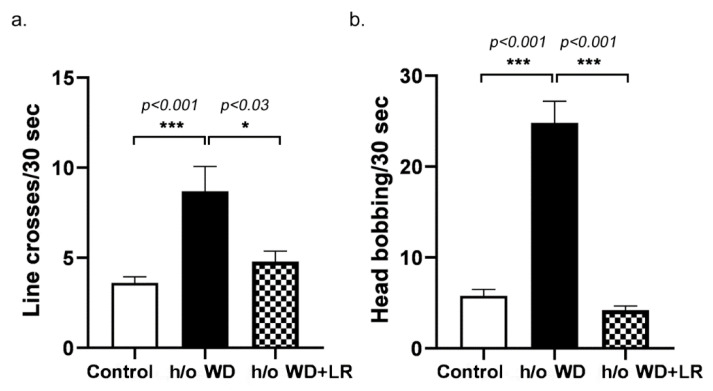
Hyperactivity (**a**) and stereotypic head bobbing (**b**) in mice with FXS-like phenotype. To examine other classic features of FXS including hyperactivity and head-bobbing stereotypy, we examined video footage of sham control animals (*n* = 6), animals with a history of (h/o) WD (*n* = 6), and animals with h/o WD + LR (*n* = 6). The animals were measured at 30 s intervals in home cages under standardized conditions. Significant differences were found between treatment groups (* *p* < 0.05 and *** *p* < 0.001).

**Figure 4 genes-13-01409-f004:**
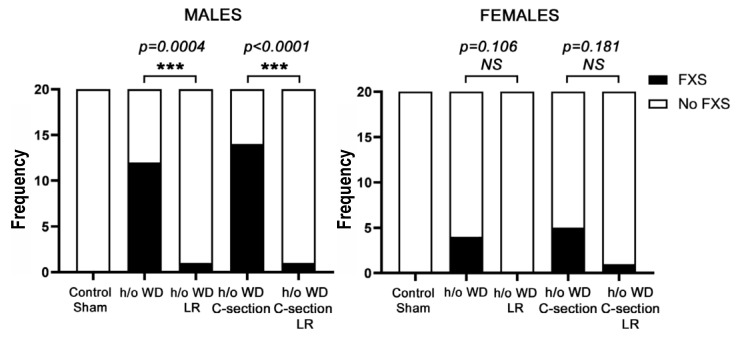
In utero probiotic *L. reuteri* effects on FXS-like phenotypes in progeny mice. To test our microbe-driven hypothesis, pregnant mothers with a history of (h/o) WD were randomly subdivided with half receiving probiotic *L. reuteri* in their drinking water (*n* = 6) and half receiving regular drinking water (*n* = 6). The frequency of FXS-like features was measured in each treatment group. Significant differences (*** *p* < 0.001) were found after in utero dosing with *L. reuteri*, and the benefits of in utero *L. reuteri* were preserved after C-section rederivation. There were no significant (NS) differences between groups in females.

**Figure 5 genes-13-01409-f005:**
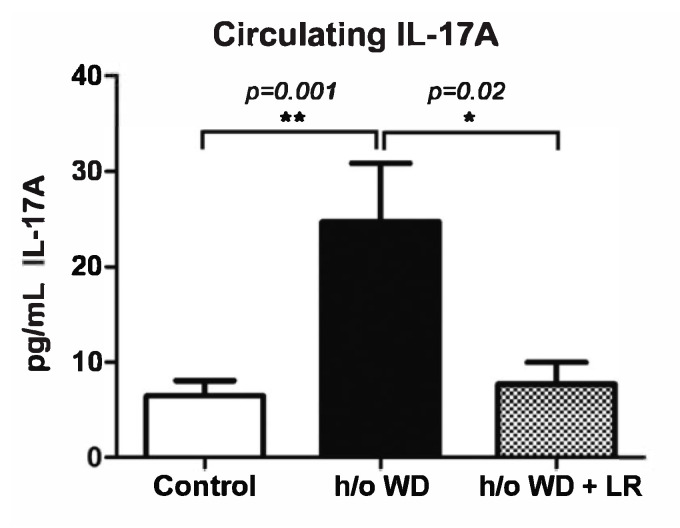
Expression of pro-inflammatory cytokine Il-17A in pregnant mother mice. To test systemic levels of inflammatory cytokines under different dietary and microbial conditions, we used whole blood from control mothers (*n* = 8), the mothers with a history of (h/o) WD (*n* = 8), and mothers with h/o WD + LR (*n* = 8).To test systemic levels of Il-17A in offspring mice, we used whole blood collected by terminal cardiac puncture and diluted 1:1. Circulating Il-17A levels were determined using ELISA at Eve Technologies (Calgary, AB, Canada). We found significant differences among treatment groups (* *p* < 0.05 and ** *p* < 0.001).

**Figure 6 genes-13-01409-f006:**
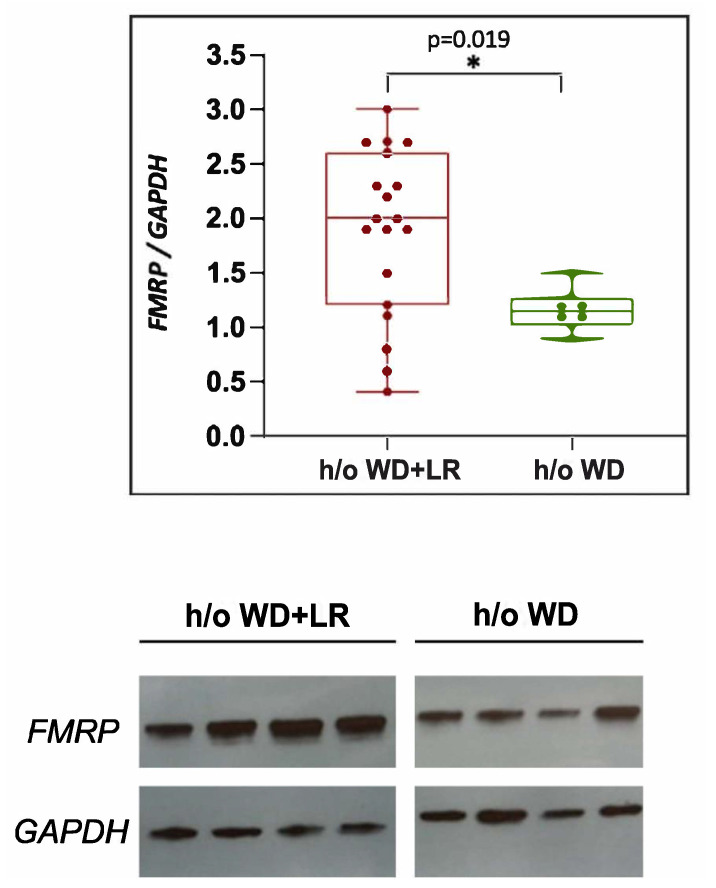
Brain expression levels of FMRP in offspring mice. FMRP expression level was measured using Western Blot analysis in male mice with a history of (h/o) WD + LR (*n* = 19) and male mice with h/o WD only (*n* = 8). Significant differences between the groups (* *p* < 0.05) were observed.

## Data Availability

Not applicable.

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
