# Peer review of "Maternal Microbiota Modulate a Fragile X-like Syndrome in Offspring Mice"

_genes, 2022, doi:10.3390/genes13081409_

Round 1
Reviewer 1 Report
First of all, I would like to thank the editor and the authors for allowing me to review this article.
The authors showed the Fragile X syndrome like behavior (FXLS) with increase of pro-inflammatory cytokine IL-17 in progeny of mice fed a Western diet and the supplementation of Lactobacillus reuteri was sufficient to inhibit FXS-like phenotypes. The FXLS was not canceled in offspring delivered by Caesarian showing the crucial role in utero contributions. The authors concluded that administration of L. reuteri during pregnancy with a WD effectively neutralized the symptoms of the neurological syndrome after the birth of infants, raising the possibility of similar therapeutic strategies in humans. The experiments are well organized, and the theme is interesting. It will be useful for readers. To make this paper better, I have a few comments.
Major comment
1. It is very interesting to note that the maternal dysbiosis in pregnancy affects the offspring behavior. However, the authors cited to previous reports without showing present microbiome data. Even if the same model was used again, I think it is important to confirm dysbiosis by showing microbiome data.
2. I agree the supplementation of L. reuteri during pregnancy ameliorated the neurodevelopmental disorder in offspring via suppressing IL-17. I wonder whether L. reuteri affect the shape of microbiome in offspring. If L. reuteri supplementation does not affect offspring’s microbiome, will F3 generation show neurodevelopmental disorder? Since the authors wrote the microbiome analysis in “Materials and Methods”, the microbiome data in each group can be shown and discussed the differences.
Minor comment
Overall
3. The figure legends should be more detailed for readers. All definitions should be included in figure legends. For example, what does the error bar mean? sd or se?
4. I recommend the authors to unify the figures for better understanding of readers. For example, the group names are different in each figure.
5. In L145 Microbiome analysis, it is recommended to add the database information for OTU annotation for reproducible.
6. In L247, the authors wrote, “We found significant differences between the Fragile X syndrome-like phenotype (FXLS) mice and control mice in ear pinnae morphology”. However, significant differences between Control and h/o WD do not seem to be shown in Figure 2.
7. In L327 Fig. 6, the expression of FMRP in h/o WD + LR group was highly variable. If the levels of FMRP relate to the behavior, the author can add the correlation analysis in each group.
Reviewer 2 Report
This study aimed to test the hypothesis that gut dysbiosis and inflammation during pregnancy influenced the prenatal uterine environment, leading to FXS-like phenotypes in offspring. They found that progeny of mice fed a Westernized diet (WD) with low fiber and extra fat exhibited higher frequencies of stereotypy, hyperactivity, cranial features and lower FMRP protein expression similar to what is typically observed in Fragile X syndrome (FXS) in humans. They also observed that oral in utero supplementation with a beneficial anti-inflammatory probiotic microbe, Lactobacillus reuteri, was sufficient to inhibit FXS-like phenotypes in offspring mice, and foster-reared offspring still displayed a high frequency of FXS-like features, while matched foster-reared progeny of L. reuteri-treated mothers did not, indicating significant in utero contributions. Furthermore, the authors found that the cytokine, IL-17, were increased in pregnant WD mice relative to matched gravid mice and to those given supplementary L. reuteri probiotic, and the inflammatory condition may modulate the methylation of FMRP. Thus, their findings suggest that diet-induced dysbiosis in the prenatal uterine environment is strongly associated with the incidence of the FXS-like phenotype in progeny but can be alleviated by addressing gut dysbiosis through probiotic supplementation. This study is interesting and novel, which may present a novel animal model for FXS, beside the classical transgenic animal models.
Strengths:
The experimental design is rigorous, especially the c-section groups, and the data is presented clearly. The statistical analysis is good and test methods were chosen correctly. The finding is interesting.
Weakness:
- As known to all, fmr1 knockout mice are classical animal model of FXS, is there any similarity in the cytokine profiles of the pregnant WD mice and the fmr1 knockout mice? There should be at least some references and discussion about this.
- Does the Lactobacillus reuteri show any rescuing effects in fmr1 knockout mice?
- In some groups, the animal numbers are 5 or 6, which is kind of small for any animal behavior tests. If possible, the authors may need to increase these animal numbers.
Round 2
Reviewer 2 Report
The authors' reply are good enough, and I think the manuscript is fine to be published.